# Exploring Format Consistency for Instruction Tuning

**Shihao Liang**[*]                                                         *shihaoliang0828@gmail.com*
*Department of Computer Science*
*Tsinghua University*

**Runchu Tian**[*]                                                         *trc20@mails.tsinghua.edu.cn*
*Department of Computer Science*
*Tsinghua University*

**Kunlun Zhu**[*]                                                         *zhuklun@mail2.sysu.edu.cn*
*Department of Computer Science*
*Tsinghua University*

**Yujia Qin**                                                         *qyj20@mails.tsinghua.edu.cn*
*Department of Computer Science*
*Tsinghua University*

**Huadong Wang**                                                         *huadw2012@163.com*
*ModelBest Inc.*

**Xin Cong**                                                         *congxin1995@tsinghua.edu.cn*
*Department of Computer Science*
*Tsinghua University*

**Zhiyuan Liu**[†]                                                         *liuzy@tsinghua.edu.cn*
*Department of Computer Science*
*Tsinghua University*

**Xiaojiang Liu**                                                         *xiaojiang_liu@apple.com*
*Apple*

**Maosong Sun**[†]                                                         *sms@tsinghua.edu.cn*
*Department of Computer Science*
*Tsinghua University*

**Reviewed on OpenReview:** *https://openreview.net/forum?id=n8fZ6mY6PB*

## Abstract

Instruction tuning has emerged as a promising approach to enhancing large language models in following human instructions. It is shown that increasing the diversity and number of instructions in the training data can consistently enhance generalization performance, which facilitates a recent endeavor to collect various instructions and integrate existing instruction tuning datasets into larger collections. However, different users have their unique ways of expressing instructions, and there often exist variations across different datasets in the instruction styles and formats, i.e., format inconsistency. In this work, we propose a framework named "Unified Instruction Tuning" (UIT), which calls OpenAI APIs for automatic format transfer among different instruction tuning datasets such as PromptSource, FLAN and CrossFit. With the framework, we (1) demonstrate the necessity of maintaining

---

[*] Indicates equal contribution.
[†] Corresponding author.

format consistency in instruction tuning; (2) improve the generalization performance on unseen instructions on T5-LM-xl; (3) provide a novel perplexity-based denoising method to reduce the noise of automatic format transfer to make the UIT framework more practical and a smaller offline model based on GPT-J that achieves comparable format transfer capability to OpenAI APIs to reduce costs in practice. Further analysis regarding variations of targeted formats and other effects is intended. The code and trained models are publicly available at `https://github.com/thunlp/UnifiedInstructionTuning`.

# 1 Introduction

Recently, instruction tuning has gained considerable attention as a potent strategy for enhancing large language models (LLMs) in following human instructions and generating appropriate responses. For instance, by reformulating various NLP tasks with an instruction template, models trained on the converted dataset exhibit powerful capabilities of zero-shot generalization on unseen tasks (Wei et al., 2021). Later studies have demonstrated that instruction tuning is critical to facilitating LLMs in grounding their inner knowledge to diverse real-world scenarios (Ouyang et al., 2022; Iyer et al., 2022; Chung et al., 2022; Ding et al., 2023). Up to now, considerable efforts have been dedicated to creating datasets for instruction tuning (Honovich et al., 2022a; Bach et al., 2022; Wei et al., 2021; Wang et al., 2022b;a; Aribandi et al., 2022) and researchers find that increasing the task diversity (i.e., the number of unique tasks) of the training data can consistently enhance generalization performance (Wang et al., 2022b; Iyer et al., 2022; Longpre et al., 2023). Therefore, the community has witnessed a growing endeavor to collect various instructions and integrate existing instruction tuning datasets into larger collections (Iyer et al., 2022; Longpre et al., 2023; Chung et al., 2022; Zhou et al., 2023).

While previous works strive to increase **task diversity** and merge existing instruction tuning datasets, they typically ignore the **format consistency** among these datasets. More specifically, different users have their unique ways of expressing instructions, even if these instructions correspond to the same intent. Hence, there often exist variations across different datasets in the instruction styles and formats, which is dubbed as the format inconsistency issue. Take the case of a summarization task, the instruction can be as detailed as "*In this task, you are given a conversation, and your task is to generate a summary... Input: ... Output: ...*" in Ni-v2 (Wang et al., 2022b) or simply composed of a few keywords, e.g., "*Summarize: ...*" in CrossFit (Ye et al., 2021b). Due to the format inconsistency issue, fine-tuned LLMs may have difficulty in handling unseen instructions in a different format at the test time, exhibiting poor out-of-distribution (OOD) generalization. Hence, before directly merging diverse datasets from various sources and performing multi-task training (i.e., the common practice), it is essential to conduct a comprehensive study of how format inconsistency may impact the performance of instruction tuning and whether mitigating such inconsistency could enhance the generalization.

However, unifying the format across different datasets is not easy. First, instructions are inherently diverse and nuanced, and the vast range of possible expressions makes it challenging to devise a fixed rule for format transfer. Second, standardizing formats can sometimes inadvertently change the meaning of the original instructions. This is particularly problematic for complex tasks where the instruction's wording and style are crucial to correctly guiding the model behavior. In this paper, we introduce a format transfer framework, **Unified Instruction Tuning (UIT)** (Figure 1) to explore the effects of format consistency. Specifically, we use OpenAI GPT3.5[1] for automatic instruction format transfer. Leveraging its powerful in-context learning capabilities, GPT3.5 can successfully transfer the instruction from a source format to a target format based on only a few handcrafted examples. Then we analyze how format inconsistency could affect generalization under two settings: (1) testing-time setting, which simulates the format inconsistency between the training data and the testing data, and (2) training-time setting, which simulates the format inconsistency among different sources of instructions in the training data. We perform analysis across five benchmarks and show that our method successfully mitigates the format inconsistency issue and improves the generalization performance on unseen instructions in both settings.

---

[1] `https://platform.openai.com/docs/models/gpt-3-5`

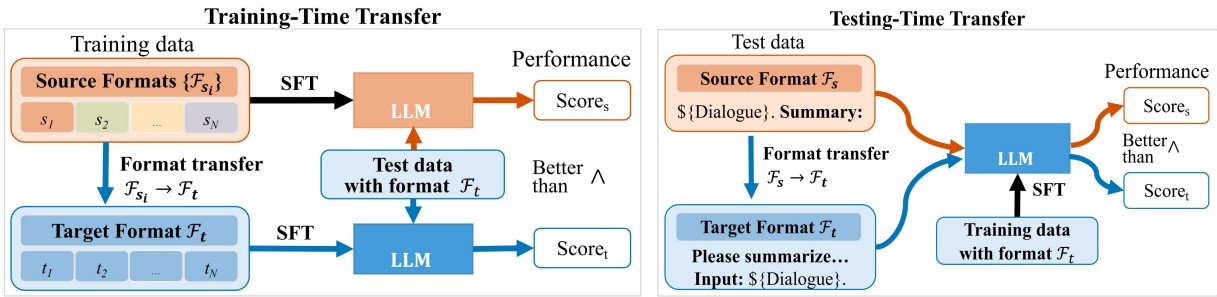

Figure 1: The proposed format transfer framework is applied to two settings: testing-time transfer and training-time transfer. $s_1, \cdots, s_N$ denote the training data in the original instruction format, $t_1, \cdots, t_N$ denote all the transferred training data in target format.

Despite its simplicity and performance, the above framework encounters two practical challenges. To begin with, the converted instructions are not as perfect as human-written ones and sometimes involve noise. For instance, an auto-converted instruction may express a slightly different meaning than the original one. To address this issue, we propose a novel perplexity-based denoising strategy that samples multiple possible conversions of a source instruction and then filters those low-quality ones based on perplexity. Experimental results reveal that this strategy effectively reduces the noise of format transfer and improves robustness and performance. Second, converting large-scale instructions via OpenAI API can result in substantial costs for API calls, which is infeasible in practice. To this end, we propose to learn an offline model for format transfer by distilling from GPT3.5. We demonstrate that with a few examples generated by GPT3.5, a much smaller model can be trained to achieve almost equivalent performance in format transfer, which saves the costs for API calls in practice. In general, our findings shed light on an essential but previously overlooked aspect, i.e., format consistency, for instruction tuning. We envision our research could inspire more efforts in advancing the instruction tuning methodologies for LLMs.

Table 1: A comparison of representative instruction tuning datasets of different instruction formats. "Num.", "Cate.", "Exp.", "Inst.", "Unnat-Inst", refer to Number, Category, Example, Instruction, and unnatural-instructions respectively.

| Resource | Task Num. | Cate. Num. | Total Exp. | Inst. format |
|---|---|---|---|---|
| **Ni-v2** (Wang et al., 2022b) | 1616 | 76 | 5M | task-level |
| **Flan 2021** (Wei et al., 2021) | 62 | 12 | 4.4M | instance-level |
| **CrossFit** (Ye et al., 2021a) | 159 | 13 | 7.1M | keywords-level |
| **P3** (Bach et al., 2022) | 62 | 13 | 12M | instance-level |
| **Unnat-Inst** (Honovich et al., 2022a) | 117 | — | 64k | task-level |
| **OPT-IML** (Iyer et al., 2022) | 1545 | 93 | 17.9M | mixed |
| **Flan 2022** (Longpre et al., 2023) | 1836 | 162 | 15M | mixed |

## 2 Related Work

**Instruction Tuning**  Instruction tuning regulates LLMs to accurately comprehend and interpret natural language instructions. Prior works in this field focus on reformulating NLP tasks using the templates of instructions. Wei et al. (2021) pioneered to show that fine-tuning LLMs on large collections of tasks formatted in instructions enables the model to generalize to unseen tasks in a zero-shot manner. Since then, there has been a surge of interest in manually constructing high-quality instruction datasets by first reformulating the formats of existing NLP datasets and then merging them (Mishra et al., 2022; Bach et al., 2022; Ye et al., 2021b; Ouyang et al., 2022). Another line of study (Longpre et al., 2023; Iyer et al., 2022) demonstrates

that scaling the number of training tasks and task diversity can further enhance the model's generalization performance. However, all these works directly mix all the existing instruction datasets while ignoring the potential issue of format inconsistency. Instead of investigating the number and diversity of training instructions, we instead explore an under-explored facet, i.e., the instruction format of instruction tuning, and investigate its impact on generalization.

**Data Augmentation**  Besides manually curating instruction tuning datasets, Honovich et al. (2022a) show that fine-tuning LLMs with machine-generated instruction tuning data achieves excellent performance compared with human-written data, indicating that data augmentation is an effective method to enhance the data quantity and task diversity, which overcomes the time-consuming issues of human annotation. Recently, Taori et al. (2023); Peng et al. (2023); Ding et al. (2023) adopt machine-annotation method (Wang et al., 2022a) to generate real-world human instructions (rather than instructions that describe NLP tasks) and model responses based on powerful LLMs such as ChatGPT. Similarly, in this paper, we also leverage LLMs for automatic format transfer and data augmentation. Since real-world instructions are quite diverse and hard to annotate their formats, we instead focus on instructions that describe NLP tasks to rigorously study the effects of instruction format. We believe the derived findings can potentially be applied to real-world instructions in the future.

**Synthetic Data Denoising**  Generative models are commonly utilized for data augmentation (Taori et al., 2023). However, these synthetic datasets are not always as reliable as those human-annotated ones, and filtering out noisy examples can boost the model performance (Le Bras et al., 2020). Recent studies have suggested different approaches for denoising. For instance, Yang et al. (2020); Fang et al. (2022) adopted influence functions (Koh & Liang, 2017) to evaluate the quality of the synthetic data; Wang et al. (2022c) employ the NLU Consistency Filtering (Anaby-Tavor et al., 2020) to filter out low-quality samples. In our research, we utilized LLMs for instruction format transfer, which may introduce noise throughout the process. To overcome this challenge, we adopted a simple and effective perplexity scoring strategy to denoise our auto-constructed dataset (section 5).

## 3   Instruction Format Inconsistency

As outlined in  Iyer et al. (2022), existing instruction formats exhibit variations across different datasets, which can be classified into three distinct hierarchical levels: Task-level format, Instance-level format, and Keywords-level format (as illustrated in Figure 2). We present an overview of existing instruction tuning datasets based on instruction formats in Table 1.

- **Task-level Format** encompasses a comprehensive definition of a task and may include supplementary information such as positive or negative examples and explanations of the examples. Representative datasets are Ni-v2 (Wang et al., 2022b), Unnatural Instructions  (Honovich et al., 2022a), and Alpaca (Taori et al., 2023).

- **Instance-level Format** employs succinct templates that are customized for each individual example and is occasionally structured in a cloze-style format to elicit the intended output. Representative datasets are Flan (Wei et al., 2021) and PromptSource (Bach et al., 2022).

- **Keywords-level Format** closely resembles the instance-level format, but it limits the instruction templates exclusively to keywords. CrossFit (Ye et al., 2021b) serves as a representative example of a keywords-level dataset.

Compared with task diversity, the effect of format consistency is poorly understood in instruction tuning. We contend that successful instruction understanding and generalization are influenced by both task diversity and format consistency. Task diversity can be enhanced by incorporating as many tasks into the training data (e.g., merging existing instruction tuning datasets) as possible. However, it is crucial to note that when merging different datasets for training, the training data originating from different sources often present variations in the instruction formats. When confronted with instructions of unseen inconsistent formats

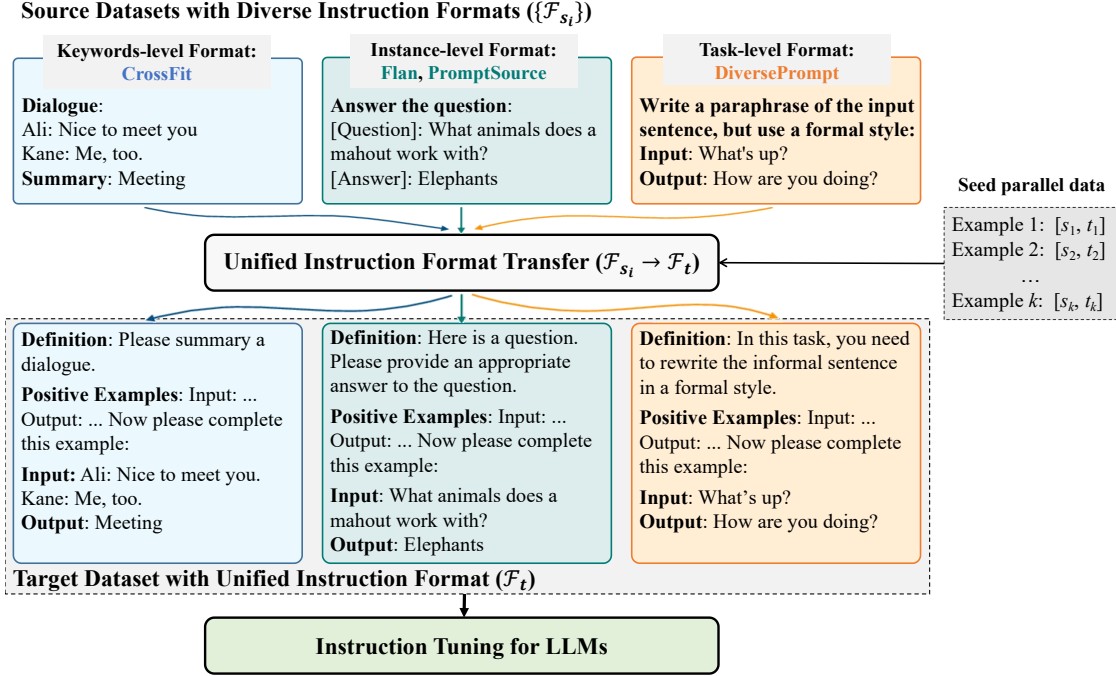

Figure 2: Transferring instruction formats with UIT. The existing instruction formats exhibit variations across different datasets, which can be classified into three distinct hierarchical formats: Task level, Instance level, and Keywords level. UIT leverages seed parallel data to conduct format transfer across different formats automatically.

at the test time, the trained model may fail to generalize well and comprehend the intent behind different instructions, showing poor OOD generalization.

## 4 Framework and Experiments

To mitigate format inconsistency, we propose a format transfer framework, *Unified Instruction Tuning* (UIT), to convert the instruction formats of existing datasets into a unified format.

### 4.1 Unified Instruction Format Transfer

Denote the target unified instruction format as $\mathcal{F}_t$ and the original instruction format of a source dataset as $\mathcal{F}_s$, we aim to convert $\mathcal{F}_s$ into $\mathcal{F}_t$ to alleviate the OOD generalization in the instruction format. Taking inspiration from Honovich et al. (2022a), we rely on the LLM's in-context learning ability to conduct format transfer in an automatic manner. Specifically, we manually select $k$ seed parallel data $\{[s_1, t_1], \cdots, [s_k, t_k]\}$, where $s_i$ and $t_i$ are the same instance (task) expressed in format $\mathcal{F}_s$ and $\mathcal{F}_t$ respectively.

Given a new instance $s_{new}$ with format $\mathcal{F}_s$, we transfer its instruction format into the unified instruction format $\mathcal{F}_t$ via in-context learning as follows:

$$t_{new} = \mathrm{LLM}\left(s_{new}, [s_1, t_1], \cdots, [s_k, t_k]\right), \tag{1}$$

where $t_{new}$ refers to the transferred instance with $\mathcal{F}_t$. We choose `text-davinci-003` (GPT3.5) as the LLM for format transfer. Details of the prompt for format transfer are shown in Figure 3.

### 4.2 Experiments

**Settings**  To simulate the format inconsistency problem, we design two experimental settings:

**Example 1**
**Source Instruction**: Translate to Spanish
**Target Instruction**: In this task, you are given a sentence in the English language. Your job is to translate the English sentence into the Spanish language.

**Example 2**
**Source Instruction**: output the opposite of
**Target Instruction**: Given an adjective, generate its antonym. An antonym of a word is a word opposite in meaning to it.

**Example 3**
**Source Instruction**: Output whether the sentiment of the input sentence is positive or negative.
**Target Instruction**: Given a review, you need to predict whether the review is good or bad. A negative review is a bad review, and positive/neutral reviews are good reviews.

**Example 4**
**Source Instruction**: Rate from 0 (definitely not) to 5 (perfectly) the degree in which both sentences describe the same event.

**Target Instruction**: Given two sentences related to a specific event or scenario, assign a score from 0 (definitely not) to 5 (perfectly) to indicate the degree in which both sentences describe the same event.

Figure 3: An example of format transfer using GPT3.5, where we prompt the model with 3 parallel examples to generate the target instruction for the 4-th example.

- **Testing-time Format Transfer**: the training data is formatted in $\mathcal{F}_t$, while the test data is formatted in $\mathcal{F}_s$. To mitigate the format inconsistency, we convert the instruction format of the test data into $\mathcal{F}_t$, without modifying the training data. This setting is designed to explore the format inconsistency impact between training data and the test data in the inference phase.

- **Training-time Format Transfer**: the training data is mixed with different formats (e.g., both $\mathcal{F}_s$ and $\mathcal{F}_t$), and the testing data is in the format of $\mathcal{F}_t$. Instead of modifying the testing data, here we convert the training data from format $\mathcal{F}_s$ to $\mathcal{F}_t$. This setting is designed to simulate the format inconsistency of different sources of the training data.

For both settings, we choose T5-LM-xl [2] as our model and use Exact Match (EM) and Rouge-L as evaluation metrics.

**Datasets**   For the testing-time setting, we select Ni-v2 (Wang et al., 2022b) as the training dataset and use DiversePrompt (Honovich et al., 2022b), Flan (Wei et al., 2021), CrossFit (Ye et al., 2021a), and PromptSource (Bach et al., 2022) as the test dataset. We evaluate the tasks that do not appear in the training stage. These tasks are the same as or similar to those in Ni-v2 test set. In Ni-v2, the instruction format incorporates four components: (1) task definition (**D**), (2) positive example (**P**) for demonstration instances with the ground-truth label, (3) negative examples (**N**) for demonstration instances with a false label, and (4) explanations (**E**) that provide detailed explanations for the examples. Different formats refer to distinct combinations of the above components. For example, the **DP** format includes the task definition and positive examples information. In our experiments, we consider four primary formats, namely **DP**, **DPN**, **DPE**, and **DPNE** as the unified instruction format, respectively.

For the training-time setting, we use the training set of Ni-v2 together with Flan, CrossFit, and P3 respectively for training and use the test set of Ni-v2 for evaluation. As Flan, CrossFit, and P3 may contain instances that exist in the test set of Ni-v2, to prevent data leakage, we filter the overlapped data in Flan, CrossFit, and P3 and use the remaining data for training. In this setting, we choose **DP** as the unified instruction format.

**Baselines**   We construct two baselines: (1) **Raw** does not involve any modifications on the instruction format for both training and testing. For instance, we directly test an instance from Flan in its original format using a model trained with Ni-v2 in **DPN** format. (2) **Heuristic** applies manually-designed rules to transfer different instruction formats into the unified one. If the information from the original format matches the corresponding field in the unified format, we fill the unified format with that information. Otherwise, we

---

[2]https://huggingface.co/google/t5-xl-lm-adapt

leave the respective field in the unified format blank. For instance, an instance from Flan can be transferred to the **DPE** format by leaving the *Definition* and *Explanations* fields blank and filling the *Positive Examples* field with randomly selected instances from the Flan training set.

Table 2: Testing-time format transfer experiment with four target unified instruction formats (**DP**, **DPE**, **DPN**, **DPNE**), respectively. We evaluate three methods: (1) raw instructions, transferred instructions based on (2) heuristic rules and (3) our proposed UIT. The training is conducted on Ni-v2 while the testing is conducted on DiversePrompt, FLAN, CrossFit, and PromptSource, respectively.

| Format | Method | DiversePrompt | | FLAN | | CrossFit | | PromptSource | | Average | |
|---|---|---|---|---|---|---|---|---|---|---|---|
| | | EM | Rouge-L | EM | Rouge-L | EM | Rouge-L | EM | Rouge-L | EM | Rouge-L |
| DP | raw | 0.1 | 4.7 | 11.6 | 20.8 | 0.2 | 3.9 | 6.6 | 13.6 | 4.6 | 10.8 |
| | heuristic | **34.7** | 45.1 | 31.4 | 44.8 | 43.7 | 56.0 | 27.3 | 32.7 | 34.3 | 44.6 |
| | unified | 34.2 | **45.4** | **32.6** | **46.3** | **49.1** | **60.1** | **29.2** | **34.7** | **36.3** | **46.6** |
| DPE | raw | 0.1 | 5.0 | 18.1 | 27.8 | 0.3 | 4.4 | 14.4 | 19.2 | 8.2 | 14.1 |
| | heuristic | 32.5 | 43.4 | 32.0 | 45.3 | 41.3 | 54.2 | 26.6 | 31.1 | 33.1 | 43.8 |
| | unified | **32.9** | **44.8** | **33.5** | **46.9** | **46.9** | **58.4** | **27.8** | **32.8** | **35.5** | **46.0** |
| DPN | raw | 0.2 | 5.5 | 12.0 | 22.6 | 0.2 | 4.5 | 5.3 | 11.6 | 4.4 | 11.1 |
| | heuristic | 30.6 | 43.5 | 31.9 | 45.3 | 43.5 | 55.5 | 29.0 | 33.9 | 33.9 | 44.8 |
| | unified | **31.5** | **44.3** | **34.8** | **48.3** | **50.3** | **60.4** | **32.4** | **38.3** | **37.5** | **48.2** |
| DPNE | raw | 0.1 | 5.2 | 15.2 | 25.3 | 0.2 | 3.8 | 19.2 | 23.7 | 8.7 | 14.5 |
| | heuristic | 30.6 | 43.4 | 30.7 | 43.6 | 42.8 | 54.6 | 29.1 | 33.7 | 33.4 | 44.1 |
| | unified | **32.2** | **43.4** | **35.0** | **48.0** | **48.6** | **59.3** | **29.8** | **34.9** | **36.6** | **46.7** |

**Results and Analyses** Testing-time format transfer results are shown in Table 2, and we find that: (1) transferring the instruction format either through the heuristic rule or our UIT significantly improves the performance than the vanilla baseline (i.e., raw), demonstrating the necessity of maintaining format consistency in instruction tuning; (2) our UIT consistently outperforms the heuristic method across all benchmarks and almost all formats in Ni-v2. Compared with the heuristic method, UIT fully utilizes the semantic understanding and generation abilities of GPT3.5 to derive better transferred instructions; (3) the **DPN** format demonstrates the highest average performance and exhibits the largest improvements with UIT.

Training-time format transfer results are shown in Table 3, which shows that format transfer also brings performance improvements compared to raw baseline and performs slightly better than the heuristic method. This again demonstrates that UIT can improve the generalization performance by unifying the instruction format. However, the improvements in the training-time setting are not as significant as those in the testing-time setting. We conjecture this may be because the format inconsistency issue is more evident in our testing-time setting than in the training-time setting. Overall, the results under both settings validate our hypothesis that mitigating the instruction format conduces to improved generalization.

**Limitations in Practice** Despite the favorable performance, the proposed framework still has some limitations: first, automatic format transfer sometimes involves noise or even errors in the generated data, which may produce adverse effects; second, the proposed method heavily relies on OpenAI API calls, which entail substantial costs especially for large-scale instruction datasets. Both issues would limit UIT's real-world deployment. In the following, we discuss potential solutions for the above limitations by proposing a denoising strategy (section 5) and training an offline transfer model (section 6), respectively.

## 5 Denoising for Format Transfer

Empirically, transferring format via LLMs will introduce noise unavoidably. The transferred instances may contain errors like critical changes to task definition or hallucinatory restrictions. Intuitively, utilizing erroneous instructions would impair the model's generalization performance. To this end, we propose a perplexity-based denoising strategy to filter low-quality instances.

Table 3: Training-time format transfer experiment with **DP** format. We compare our UIT with two baselines: raw instructions and instructions transferred by the heuristic rule. The training dataset is Ni-v2 combined with CrossFit, Flan, or P3.

| Method | +CrossFit | | +FLAN | | +P3 | | Average | |
|---|---|---|---|---|---|---|---|---|
| | EM | Rouge-L | EM | Rouge-L | EM | Rouge-L | EM | Rouge-L |
| raw | 37.5 | **56.0** | 38.4 | 56.9 | 38.8 | 56.7 | 38.2 | 56.5 |
| heuristic | 37.7 | 55.9 | **38.9** | 57.4 | **39.9** | **58.1** | **38.8** | **57.1** |
| unified | **37.9** | **56.0** | **38.9** | **57.5** | 39.4 | 57.3 | 38.7 | 56.9 |

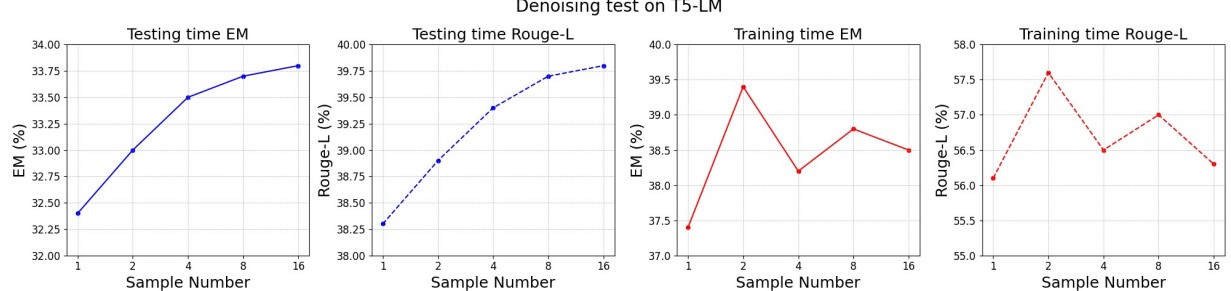

Figure 4: The performance of the denoising strategy at the testing and training time with different number of samples. Detailed results in the form of a table are presented in Section B of the appendices.

**Perplexity-based Denoising Strategy**  Perplexity (PPL) is a widely used metric for evaluating the semantic coherence and certainty of language models. We assume that noisy instructions can reduce the certainty of LLMs in accurately predicting the correct output token sequence[3], leading to higher perplexity. As a result, perplexity serves as a useful metric for assessing the quality of transferred instructions. Hence, we propose to sample multiple times from LLM to obtain multiple transferred instructions. Then we calculate the perplexity for each instruction. Specifically, we concatenate the transferred instruction and the input query, then predict the annotated label and calculate its perplexity, and filter those with high perplexity.

We employ GPT3.5 with temperature 1.0 to perform sampling for $N$ times with different random seeds, where $N$ is chosen from $\{1, 2, 4, 8, 16, 32\}$. Then, we sort the generated instructions based on perplexity using GPT-J (Wang & Komatsuzaki, 2021) and select the sample with the lowest perplexity. We compare our method with the baseline that only samples once from GPT3.5. We conduct experiments on Ni-v2 and PromptSource under both the testing-time and training-time settings. For the former, we select the transferred instruction samples with the lowest perplexity; while for the latter, we incorporate multiple transferred results with lower perplexity as the training data.

**Results**  As shown in figure 4, our proposed denoising strategy stably improves the performance at the testing time, and this improvement continues to increase when more instructions are sampled, which shows our method can successfully filter out those low-quality instructions to reduce noise during format transfer. In addition, the method can also improve performance in the training-time setting but the improvement is not more evident when more transferred instructions are included in the training data. It reveals that the model is less sensitive to noise during the training phase.

## 6   Training Offline Model for Format Transfer

Converting large-scale instructions via OpenAI API can cause substantial costs for API calls. To alleviate the reliance on OpenAI API, it is necessary to derive an offline model that has comparable format transfer

---

[3]We merely use the positive example (**P**) as mentioned in section 4.2 for noise assessment.

performance to GPT3.5 but involves fewer costs. Hence we propose to distill the format transfer ability of GPT3.5 into small-scale models.

**Fine-tuned Offline Model with Knowledge Distillation**   Compared with larger models, small offline models are less capable of completing format transfer directly through in-context learning without training. Therefore, we strive to enhance small-scale models via knowledge distillation (Hinton et al., 2015). In pursuit of higher quality, we always make GPT3.5 convert the relatively complex and informative instruction format (e.g., Ni-v2) into a simpler and less informative one (e.g., PromptSource). In this way, we obtain parallel data and use it to fine-tune GPT-J for format transfer. We use the generated PromptSource-style instructions as the source and the original Ni-v2 instructions as the target to construct a dataset of approximately 3,000 instances. To assess the quality of GPT-J's transfer results, we compare them with the heuristic baseline and GPT3.5's conversion results in the testing-time setting with two formats (**DP** and **DPN**).

Table 4:  Results of training an offline model (GPT-J) for format transfer at testing time. We compare the transferred instructions using heuristic rules, GPT3.5, or our fine-tuned GPT-J. Other settings are similar to those in Table 2.

| Format | Method | DiversePrompt | | Flan | | CrossFit | | PromptSource | | Average | |
|---|---|---|---|---|---|---|---|---|---|---|---|
| | | EM | Rouge-L | EM | Rouge-L | EM | Rouge-L | EM | Rouge-L | EM | Rouge-L |
| DP | heuristic | 34.7 | 45.1 | 31.4 | 44.8 | 43.7 | 56.0 | 27.3 | 32.7 | 34.3 | 44.6 |
| | GPT3.5 | 34.2 | 45.4 | 32.6 | 46.3 | **49.1** | **60.1** | 29.2 | 34.7 | **36.3** | **46.6** |
| | GPT-J | **35.2** | **45.6** | **33.5** | **46.6** | 43.6 | 54.5 | **31.6** | **36.4** | 36.0 | 45.8 |
| DPN | heuristic | 30.6 | 43.5 | 31.9 | 45.3 | 43.5 | 55.5 | 29.0 | 33.9 | 33.9 | 44.8 |
| | GPT3.5 | 31.5 | 44.3 | **34.8** | 48.3 | **50.3** | **60.4** | 30.8 | 36.1 | **37.1** | **47.6** |
| | GPT-J | **34.7** | **45.7** | **34.8** | **48.4** | 46.0 | 55.5 | **31.4** | **36.5** | 36.7 | 46.5 |

**Results**   As exhibited in Table 4, the fine-tuned GPT-J performs much better than the heuristic baseline but slightly worse than GPT3.5. This shows that our method can distill the format transfer ability into small-scale models, which saves the costs in practice. Additionally, the performance is highly correlated with the similarity of the source and target formats. For instance, for DiversePrompt whose instruction format is similar to the target format, the transfer process is less challenging. As a result, the fine-tuned model demonstrates comparable or even superior performance than GPT3.5. Conversely, for CrossFit which only describes keywords and lacks natural language instructions, it is more difficult for small models to produce high-quality instructions, resulting in inferior performance.

## 7   Further Analysis

**Effects of the Target Unified Format**   In previous experiments, we mainly use Ni-v2 as the target instruction format. To verify the versatility of UIT for various target instruction formats, we select Flan, an instance-level dataset as the target dataset and conduct testing-time transfer experiments. Results are shown in Figure 6, from which we find that testing-time format transfer brings even more significant performance improvements than the scenario when Ni-v2 is selected as target dataset. This again validates our hypothesis that format consistency is essential to OOD generalization for instruction tuning, no matter which target format is.

**Effects of Model Scaling**   As observed in previous works (Iyer et al., 2022), larger models tend to perform better in following human instructions. We also conduct model scaling experiments in the testing-time setting with T5 (Raffel et al., 2020), with the model size ranging from 5 million (T5-small) to 10 billion (T5-XXL). Results presented in Figure 5 demonstrate that in general, the performance tends to improve as the model size increases. These findings suggest that instruction format consistency is consistently essential to language models of various sizes.

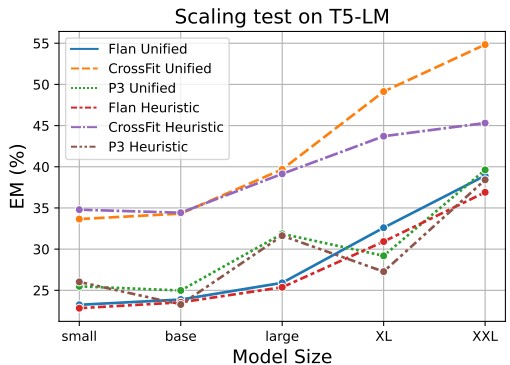

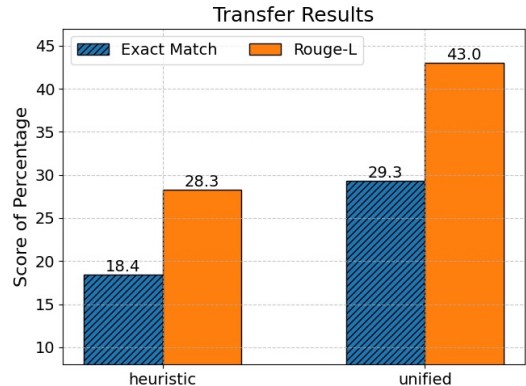

Figure 5: Results of T5-LM of different model sizes on the testing-time transfer setting.

Figure 6: Testing time transfer experiment results when Flan is selected as the target dataset and Ni-v2 as the source dataset. Transferring Ni-v2 to the target format brings significant performance improvements during inference when training is conducted with target format. Results in the form of a table are presented in Section C of the appendices.

Table 5: Experiments for task diversity and format consistency. For task diversity, we set the training dataset to `src+same`, `src+diff` or `src+same+diff`. For format consistency, we either use the raw format or use the unified format.

| Method | src+same | | src+diff | | src+same+diff | |
|---|---|---|---|---|---|---|
| | EM | Rouge-L | EM | Rouge-L | EM | Rouge-L |
| raw | 29.3 | 46.3 | 28.3 | 45.3 | 29.1 | 45.8 |
| unified | 30.8 | 47.6 | 30.7 | 47.7 | **31.0** | **47.8** |

**Task Diversity v.s. Format Consistency**   We show that both task diversity and format consistency have impacts on the generalization performance for instruction tuning. As task diversity can only be a variable during the training stage, we only conduct training-time transfer experiments. Specifically, we choose Ni-v2 as the target dataset with **DP** as target format and P3 as the source dataset. We first randomly select 20 tasks from Ni-v2 (denoted as `src`). Then we choose the same 20 training tasks from P3, denoted as `same`, and 20 different tasks from P3, which is denoted as `diff`. We treat whether to integrate `same` or `diff` to the training set (`src`) as a variable and evaluate on the original Ni-v2 test set.

As shown in Table 5, no matter which tasks are chosen as the training data, our UIT always performs better than the vanilla baseline (raw), which again demonstrates the importance of format consistency. We can also observe that without format unification, `src+same` performs better than `src+diff`, which indicates that increasing task diversity may be inferior without format consistency. Besides, `source+same+diff` with UIT performs the best among all combinations, suggesting that increasing task diversity and maintaining format consistency at the same time is the best practice for merging datasets in instruction tuning. We believe this finding can guide practitioners to better prepare the datasets for instruction tuning in the future.

## 8   Conclusion

In this paper, we propose the unified instruction-tuning framework (UIT), a standardized approach to enhancing the generalization ability for instruction tuning by unifying the format of existing instruction tuning datasets and enabling format transfer between them with LLMs like GPT-3.5. With the framework, we (1) exhibit the significance of format consistency in instruction tuning; (2) enhance the generalization

performance (9.3% in Exact Match, 7.6% in Rouge-L) on various datasets such as PromptSource, FLAN and CrossFit on T5-LM-xl; (3) propose a denoising method and an offline model training method to make our UIT more feasible in practice.

In general, we study an under-explored facet, i.e., the format consistency, for instruction tuning, and we hope our work could facilitate more attempts in relevant areas.

## 9 Limitation

While our proposed UIT framework and format transferer offer a promising approach to enhancing the generalization performance of instruction-tuned LLMs, several limitations should be acknowledged. Firstly, our method relies on the assumption that the user knows the target instruction format in advance, which may not always be the case. Secondly, we focus on instruction tuning for NLP tasks, instead of broader settings (e.g., real-world instructions (Taori et al., 2023; Chiang et al., 2023)) where formats are hard to define. We expect future works to explore whether our UIT framework can be applied to broader scenarios.

## Acknowledgement

This work is supported by the National Natural Science Foundation of China (Grant No. 62306159).

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

## Appendices

## A   Case Study

We list some examples of our format transfer process in this section. You can find examples of **D**efintion, **P**ositive examples, **N**egative examples and **E**xplanation in these cases.

---

**Transferring from P3 to Ni-v2 (Task: trec):**
**Instruction:** You are given a sentence that contains a question and a possible answer type. Your task is to identify the correct answer type from the suggested options. You may need to read the sentence and its context carefully in order to determine the correct answer type.
**Positive Examples:**
**Input:** What do bee hives do in cranberry bogs ?\nIs this asking about Description, Entity, Abbreviation, Person, Quantity, Location?
**Output:** Description
**Explanation:** The question is asking for a description of what bee hives do in cranberry bogs. So the correct answer type is Description.
**Negative Examples:**
**Input:** What golfing accessory was patented by George Grant on December 12\nIs this asking about Description, Entity, Abbreviation, Person, Quantity, Location?
**Output:** Quantity
**Explanation:** The sentence is asking about a golfing accessory. These types of questions typically require an answer about an entity (i.e. a specific object or thing), so the correct answer type is 'Entity' rather than 'Quantity'.

---

**Original P3(trec):**
**Instruction:** {Input}\n\nDescriptors: Description, Entity, Abbreviation, Person, Quantity, Location\n\nBest Descriptor?\n
**Positive Examples:**
**Input:** What do bee hives do in cranberry bogs ?\n
**Output:** \nDescription

---

**Transferring from Flan to Ni-v2 (Task: wsc):**
**Instruction:** In this task, you are given two sentences (sentence 1 and sentence 2). If sentence 1 implies that sentence 2 is true, answer "Yes", otherwise "No".
**Positive Examples:**
**Input:** sentence 1: As Ollie carried Tommy up the long winding steps, his legs ached. sentence 2: Tommy's legs ached. options: - no - yes.
**Output:** no
**Explanation:** The sentence 1 does not imply that Tommy's legs ached. So, the output should be "No".
**Negative Examples:**
**Input:** Sentence 1: Paul couldn't find his car keys, so he had to walk. Sentence 2: Paul had to walk.
**Output:** Yes
**Explanation:** Sentence 1 implies that sentence 2 is true, so the correct output should be "Yes".

---

**Original Flan(wsc):**
**Instruction:** If "{sentence1}", can we conclude that "{sentence2}"\n{options_}
**Positive Examples:**
**Input:** If "As Ollie carried Tommy up the long winding steps, his legs ached.", can we conclude that "Tommy's legs ached."\n\nOPTIONS:\n- no\n- yes.
**Output:** no

---

**Transferring from CrossFit to Ni-v2 (Task: imdb):**
**Instruction:** You are given a sentence or a paragraph describing a particular topic. Your task is to classify the sentiment of the sentence/paragraph as either negative or positive. Label the sentiment in the output as per the given parameters without incorporating any additional information into your answer.
**Positive Examples:**
**Input:** I am a big fan of Faerie Tale Theatre and I've seen them all and this is one of the best! It's funny, romantic, and a classic. I recommend this for all ages. It's great for little kids because it's well, Cinderella and great for adults and teens because it's funny and not over the top. I watched it when I was little and I still watch it now. It has great lines that my family and I quote all the time. The acting is great and it never gets old. If you like fairy tales and romances you will love this. I've watched many a Cinderella movie in my time and this is the best of them all. (Sorry Disney) I highly recommend this movie and all the Faerie Tale Theatre shows. They all appeal to all ages and are all unique and very entertaining.
**Output:** positive
**Explanation:** The sentence describes a very positive opinion on the chosen topic. The opinion is supported by facts, like the uniqueness of the show, its lasting values, great acting, and so on. Hence, the sentiment of the sentence is classified as positive.
**Negative Examples:**
**Input:** I know a few things that are worst. A few. It had a couple of funny scenes. It is a movie not appropriate for kids but, only a child would find this movie hilarious. This is definitely a movie that you would like to use a free rental coupon for. Don't waste your money just to laugh a couple of times.
**Output:** Positive
**Explanation:** The given sentence is mainly negative in nature as it suggests not to waste money on the movie. The words \"definetly\" and \"a couple of funny scenes\" are used in the sentence to provide a bit of contrast, yet it does not make the overall sentiment of the sentence positive. Therefore, the correct answer should be \"negative\" instead of \"positive\".

---

**Original CrossFit(imdb):**
**Instruction:**
**Positive Examples:**
**Input:** I am a big fan of Faerie Tale Theatre and I've seen them all and this is one of the best! It's funny, romantic, and a classic. I recommend this for all ages. It's great for little kids because it's well, Cinderella and great for adults and teens because it's funny and not over the top. I watched it when I was little and I still watch it now. It has great lines that my family and I quote all the time. The acting is great and it never gets old. If you like fairy tales and romances you will love this. I've watched many a Cinderella movie in my time and this is the best of them all. (Sorry Disney) I highly recommend this movie and all the Faerie Tale Theatre shows. They all appeal to all ages and are all unique and very entertaining.
**Output:** positive

---

**Transferring from Ni-v2 to Flan (Task: dialogre):**
**Instruction:** {input} Identify the name of one of the speakers in the given dialog.

---

**Original Ni-v2(dialogre):**
**Instruction:** You are given a dialog between 2 or more individuals. Within the dialog, there will be clues as to the names of the speakers. You will be asked at the end of the dialog to identify the name of one of the speakers.

---

# B   Detailed Results of the Denoising Strategy

This is the detailed results of the performance of the denoising strategy with different number of samples.

Table 6: The performance of the denoising strategy at the testing and training time with different numbers of samples.

| Sample Num | Testing time | | Training time | |
|---|---|---|---|---|
| | EM | Rouge-L | EM | Rouge-L |
| 1 | 32.4 | 38.3 | 37.4 | 56.1 |
| 2 | 33.0 | 38.9 | **39.4** | **57.6** |
| 4 | 33.5 | 39.4 | 38.2 | 56.5 |
| 8 | 33.7 | 39.7 | 38.8 | 57.0 |
| 16 | **33.8** | **39.8** | 38.5 | 56.3 |

## C  Results with Flan Selected as the Target Dataset

This is the detailed results of Testing time transfer experiment results when Flan is selected as the target dataset and Ni-v2 as the source dataset.

Table 7: Testing time transfer experiment results when Flan is selected as the target dataset and Ni-v2 as the source dataset. Transferring Ni-v2 to the target format brings significant performance improvements during inference when training is conducted with target format.

| Source | Target | Method | EM | Rouge-L |
|---|---|---|---|---|
| Ni-v2 | Flan | heuristic | 18.4 | 28.3 |
| Ni-v2 | Flan | unified | **29.3** | **43.0** |

## D  Seed Data

**Example 1**

**Task description A**: Review: {sentence} Is this movie review sentence negative or positive? {options_}

**Task description B**: In this task, you are given sentences from movie reviews. The task is to classify a sentence as "POS" if the sentiment of the sentence is positive or as "NEG" if the sentiment of the sentence is negative

**Positive examples**: Input b positive 1: It 's a lovely film with lovely performances by Buy and Accorsi. Output b positive 1: POS **Explanation** b positive 1: The sentiment of the sentence is positive. Hence, the label is 'POS'.

Input b positive 2: Here's yet another studio horror franchise mucking up its storyline with glitches casual fans could correct in their sleep. Output b positive 2: NEG **Explanation** b positive 2: The sentiment of the sentence is negative. Hence, the label is 'NEG'.

**Negative examples**: Input b negative 1: A smart, witty follow-up. Output b negative 1: NEG **Explanation** b negative 1: Although the sentiment of the sentence is positive, the label is 'NEG'. Hence, the label should be 'POS'.

Input b negative 2: Ultimately feels empty and unsatisfying, like swallowing a Communion wafer without the wine. Output b negative 2: POS **Explanation** b negative 2: Although the sentiment of the sentence is positive, the label is 'POS'. Hence, the label should be 'NEG'.

**Example 2**

**Task description A**: {question1} {question2} Would you say that these questions are the same? {options_}

**Task description B**: Here are two questions (Question1 and Question2). If these questions have the same meaning and same answer, answer "Yes", otherwise "No".

**Positive examples**: Input b positive 1: Question1: How do I get into my Instagram if I forgot my email and my Facebook password?, Question2: I forgot my password and also my email password. how can I get back that account? Output b positive 1: Yes **Explanation** b positive 1: These questions have the meaning and the same answer. So, the output should be "Yes".

Input b positive 2: Question1: Why don't Hong Kong residents emigrate from their cramped & stressful city, like to places such as Australia?, Question2: Why made Hong Kong so attractive to Britain as a colony given that it was the last of Britain's colonies and Britain does not profit from taxing Hong Kong? Output b positive 2: No **Explanation** b positive 2: The first question is about the emigration of Hong Kong residents and the second question is about the attraction of Hong Kong. So, they don't have the same meaning.

**Negative examples**: Input b negative 1: Question1: Why are there so many accidents on I-880?, Question2: Were there accidents in outer space? Output b negative 1: Yes **Explanation** b negative 1: Question1 asks about the cause of the accidents, while question2 inquires about their existence. So, they are different and the correct output should be "No".

Input b negative 2: Question1: How do you determine the number of neutrons of an element or its ion?, Question2: How do you find the number of neutrons in an element? What are some examples? Output b negative 2: They are the same. **Explanation** b negative 2: Note that you need to answer with "Yes" or "No" and other answers are not acceptable.

**Example 3**

**Task description A**: {context} Generate a question about the above context.

**Task description B**: Based on the given context, craft a common-sense question, especially those that are LONG, INTERESTING, and COMPLEX. The goal is to write questions that are easy for humans and hard for AI machines! To create such questions, here are some suggestions: A. What may (or may not) be the plausible reason for an event? B. What may (or may not) happen before (or after, or during) an event? C. What may (or may not) be a plausible fact about someone (or something)? D. What may (or may not) happen if an event happens (or did not happen)? You can also create other types of questions. DO NOT make your question answerable without looking at the context, or question of which the correct answer can be directly extracted from the context. DO NOT ask a question that requires very specialized knowledge that is not common sense. DO NOT ask too simple or too short questions. Your question must be related to the context and answerable with common sense. Try to add more variations and complexity to the questions.

**Positive examples**: Input b positive 1: Context: I was told, in person over the phone, that my shoes were on their way. They have my money. I have no shoes. Output b positive 1: What may happen before I called them? **Explanation** b positive 1: The question can not be answered directly from context and requires commonsense.

Input b potitive 2: Context: you see , at my age relationship is kind of important and i thought i got the one after all these years . I noticed that once again i was wrong . i was good simply because i was good , i was caring , helping , supportive , bla bla blaaa . Output b potitive 2: What may happen to me? **Explanation** b positive 2: The question can not be answered directly from context and requires commonsense.

**Negative examples**: Input b negative 1: Context: I was told, in person over the phone, that my shoes were on their way. They have my money. I have no shoes. Output b negative 1: What is on the way to my home? **Explanation** b negative 1: It can be directly answered with a span of the context and does not require any commonsense reasoning.

Input b negative 2: Context: GPS technology dates back to the time when first ever satellite was launched in the sky in 1979. The era of global positioning started then. Output b negative 2: What was launched in the sky in 1979? **Explanation** b negative 2: It can be directly answered with a span of the context and does not require any commonsense reasoning.

# E   Model Implementation Details

The hyper-parameters for training include a maximum source data length of 1024, a maximum target data length of 128, a cap of 100 instances per task for both training and evaluation, a batch size of 16 for training, a learning rate of 0.00001, a total of 2 training epochs, linear learning rate scheduling, and a warm-up period consisting of 1000 steps.

