# OpenReview forum: "Exploring Format Consistency for Instruction Tuning"
_TMLR — Accepted by TMLR_

### Review · Reviewer_HFkM · 2023-09-30

**Summary Of Contributions:**

In this paper, the authors explicitly model the concept of format consistency for instruction tuning which is important to a number of down-streamed LLM-based applications. Specifically, a format transfer framework named Unified Instruction Tuning (UIT) is developed. Together with the denoising strategy and offline model training, empirical improvements are shown in some experiments to support their efficacy.

**Audience:**

Yes

**Claims And Evidence:**

No

**Requested Changes:**

Please see the weakness section for the requested changes.

**Strengths And Weaknesses:**

**Pros:**

+ To my knowledge, this is the first work that explicitly models and considers the format consistency problem for instruction tuning. But I would like to admit that I am not an expert in this area so I cannot accurately assess the technical value of this submission.
+ The paper is well-written. I can follow it easily. And the discussions on limitations are included.
+ Comprehensive quantitative results are provided with good illustrations for concept demonstration.

After reading, I’m satisfied with most of the contents in this submission, except for some minor points:

**Cons:**

+ The proposed method looks to be ad-hoc. Specifically, the OpenAI GPT 3.5 based framework, the format denoising and the offline model training appear to be three different improvement tricks instead of one single solution based on one point. I am looking forward to a more principled solution. But I guess it is just a problem of personal taste.
+ Table 1-7 mainly report the quantitative comparisons for the ablated alternatives. I am curious whether the format consistency could help the existing instruction tuning methods. I guess the format consistency is perpendicular to their technical contribution and it is not difficult to implement? For example, what happens when the format consistency is incorporated into the state-of-the-art models which are trained over a set of datasets? Could it significantly improve the empirical results?
+ Except the reported quantitative scores, can the authors also show some qualitative comparisons? For example, can the authors show a pair of outputs where one is from the full model with the format consistency but another is not equipped with? Can the authors show some comparison results for several different SOTA baselines?

---

> ### Author Response · Authors · 2023-10-15
> **Responses to Reviewer HFkM**
>
> We appreciate the time and effort you have put into reviewing our paper. Please find our responses to your comments and concerns below:
>
>
>
>
> **Regarding Question 1:**
>
> From our perspective, denoising and the training of an offline model serve as two patches to rectify the limitations of the OpenAI GPT-3.5 based framework. They are aimed to address the following limitations of the OpenAI GPT-3.5 based framework.
>
> - In the few-shot scenario, our framework based on OpenAI GPT-3.5 does not exhibit stable performance. The introduction of denoising strategies aims to alleviate this specific shortcoming.
>
> - Our framework built upon OpenAI GPT-3.5 heavily relies on non-open-source online models. By distilling an offline open-source model using data generated by that model, we hope to make the format transfer more accessible.
>
> Thus, they provide a comprehensive solution. We will emphasize this connection in our revisions.
>
>
>
> **Regarding Question 2:**
>
> We have validated our method on existing instruction tuning datasets like FLAN, PromptSource and Ni-v2, which we consider traditional. Other more recent instruction tuning datasets like Alpaca will be considered in the future.
>
> Our primary focus remains on traditional instruction tuning datasets for two main reasons:
>
> - Their schemas and formats are more discernible and identifiable than those in real-scenario oriented datasets like Alpaca.
>
> - The evaluation techniques employed for these traditional datasets are more reliable for our analytical purposes compared to benchmarks like AlpacaEval whose evaluation relies on LLMs.
>
> We acknowledge the importance of including experiments on datasets similar to Alpaca to provide a comprehensive view. We've earmarked this for future exploration.
>
>
>
>
> **Regarding Question 3:**
>
> We've included our qualitative comparison cases in the appendix. These are also presented in the "reply to all reviewers" section.
>
>
> **May we ask if our explanation satisfies you?**

---

### Review · Reviewer_NsUB · 2023-10-10

**Summary Of Contributions:**

This paper proposes to tackle the OOD generalization of large language models on diverse datasets from the perspective of instruction format tuning. Specifically, a framework called "Unified Instruction Tuning" is introduced, which utilizes the LLM's in-context learning to perform instruction format transformation into a unified format for diverse datasets. Experiments on various benchmarks as well as different settings demonstrate the effectiveness of the proposed framework.

**Audience:**

Yes

**Broader Impact Concerns:**

None.

**Claims And Evidence:**

Yes

**Requested Changes:**

1. Please clarify the results in Table 2 by providing more details of the baseline "Heuristic" and discussing the scenario of format transformation for each test set.
2. Please discuss the cost of in-context learning and the failure cases.
3. Please include explanations (E) in Table 5 or provide the reason why it is excluded.

**Strengths And Weaknesses:**

Pros:
1. The paper is well-organized and the proposed framework is easy to follow.
2. The studied OOD generalization problem among diverse datasets can be valuable.
3. The evaluation on various benchmarks demonstrates the effectiveness of the proposed format transfer.

Cons:
1. The major contribution lies in the format transfer from source format $\mathcal{F}_s$ to a unified one $\mathcal{F}_t$. Although the authors provide several variants of unified formats in Table 2, some of the comparisons are unclear or missing. For example, how is the baseline "Heuristic" constructed? How about directly transferring the format of source $\mathcal{F}_s$ to the one in test $\mathcal{F}_t$ for each test set instead of a unified format across all the test sets?
2. The cost of LLM's in-context learning in practice as well as the ratio of failure cases are unclear.
3. The explanations (E) is excluded in Table 5.

---

> ### Author Response · Authors · 2023-10-15
> **Responses to Reviewer NsUB**
>
> Thank you for the thorough review of our manuscript. Below, we have addressed each of your comments and concerns in detail.
>
>
>
> **Regarding Question 1: Format Transformation**
>
>
>
> - **Heuristic**: Following is the definition of our heuristic method. When information from the original format aligns with a field in our proposed format, we populate that field accordingly. If not, the field remains empty. Here is an example: A task from Flan reading "You need to complete..." would be transformed into Ni-v2's DP format as:
>
>   ```
>   Task description: You need to complete...
>
>   Positive Example: None.
>   ```
>
> - **Directly transferring**: This approach mirrors what we implement during our test-time experiment. We transform dataset A to match the test set format of dataset B and then evaluate the T5-LM (fine-tuned on B's data) using the adapted A data.
>
> **Regarding Question 2: Cost and Failure Cases**
>
>
>
> - **In-Context Learning Cost**: On average, one piece of data requires approximately 900 prompt tokens and 300 response tokens. The financial cost equates to about 0.024 dollar. Therefore, unifying 1000 instances would cost approximately 24 dollars. This remains more cost-effective than employing human labor.
>
>
>
> - **Failure Cases:**   This is success rate of each part for 300 transferred instructions evaluated by 4 human annotators.
>
>   | **Item**          | **Crossfit** | **P3** | **Flan** | **Average** |
>   | :---------------- | ------------ | :----: | -------- | ------- |
>   | Definition        | 75           |   73   | 88       | 79      |
>   | Negative Examples | 54           |   51   | 21       | 42      |
>   | Explanation       | 47           |   53   | 24       | 41      |
>
> **Regarding Question 3: Exclusion of "E" in Table 5**
>
> In Table 5, we simply aim to demonstrate that the trained open-source **GPT-J** model can effectively substitute **GPT-3.5** within the framework. We believe that the performance across two different formats and four distinct datasets proves the feasibility of this replacement. Therefore, we have not included the "**E**" part.
>
> **We're grateful for your constructive feedback and will endeavor to address these points thoroughly in our revised submission.**

---

### Review · Reviewer_nGZc · 2023-10-11

**Summary Of Contributions:**

Motivated by reduced performance in out-of-distribution settings, this paper describes a method for automatically transferring datasets of one format into a dataset of a different format. The method uses either heuristics or LLMs to perform the transfer. The transfer can be applied either at training time or testing time. The paper also introduces methods for denoising the samples from the LLM and distilling potentially cheaper models from the LLM.

**Audience:**

Yes

**Broader Impact Concerns:**

None.

**Claims And Evidence:**

Yes

**Requested Changes:**

Critical:
* Provide details on how experiments were run (e.g., model details, manual intervention, heuristics) as mentioned in weaknesses.
* Summarize and clarify the idea of the work and the analysis of results as mentioned in weaknesses. Ideally, key takeaways of the paper and every figure are clear (e.g., including summary statistics or general trends).

Suggested:
* Add ablations and study on the variability of the results as mentioned in the weaknesses.
* Reorganizing and merging Section 4, 5, 6, 7 to reduce overlap and fragmentation as mentioned in the weaknesses.

**Strengths And Weaknesses:**

Strengths:
* Approach is intuitive and can likely be used broadly due by being focused on fixing the data itself.
* Sections 1-3 were easy to follow.
* Grammar and spelling seem good.

Weaknesses:
* The paper is written in a way that perhaps naturally follows in the development of research but is awkward to read, starting at Section 4. These later sections continue to expand on the base idea presented in the earlier sections, namely by focusing on the dataset dimensions D/P/N/E.
    * Section 4 claims success but Section 5 proposes mitigations to limitations in Section 4. An alternative way to structure the experiments would be to combine Section 4 and 5 together, where “raw”, “heuristic”, “unified” were compared alongside “unified_denoised”.
    * Similarly, Table 5 is overlapping data from Table 2.
    * Section 7 seems to not have a coherent story, making it hard to follow. Additionally, the experiments and corresponding results seem more limited in scope compared to the rest of the paper.
* There are key areas with missing details for how the idea of the work maps to a concrete implementation (e.g., what is the data, program, manual intervention required):
    * Can you provide more information on the training/testing setup? The only relevant details I saw were that the model is t5-LM-xl. For example, it’s not clear how pre-training, fine-tuning, and in-context learning fit into the evaluation, which is itself split between a training and testing evaluation. How does it compare to Tk-INSTRUCT in the Ni-v2 work? Section 6 similarly has more training details that could have been included (loss curves, hyper parameters, etc.).
    * Can you provide more information on the manual labeling for GPT-3.5 exemplars? I don’t see any methodology of how this is done.
    * Can you provide more information on the heuristic approach? It similarly requires manually designed rules. Furthermore, it seems quite competitive to Unified Instruction Tuning and doesn’t need the mitigations (e.g., denoising) that are required for using an LLM.
* Summary parts of the work speak in general terms but the reader will want to know what models were evaluated, what was the range of improvements, etc.
    * Abstract and conclusion don’t have any specific results
    * Nearly all the results are complex tables (e.g., data is not naturally ordered), so the reader has to compare every table entry with every other table entry to get a sense of trends. Even Tables with a natural ordering, such as the N variable in Table 4, are presented in table format rather than something like a line plot. Color, annotations with percentage improvement to a baseline, and analysis in the caption can help reduce the burden on the reader.
* It’s not clear where the approach’s generality breaks down. How specific is the framing of D/P/N/E in the experiment setup to Ni-v2? While the approach seems intuitive, the evaluation is difficult to understand because the approach is described in a general manner (before the evaluation) but the evaluation is specific to aspects of particular datasets (e.g., Ni-v2). The evaluation’s focus makes it seem like it would be easier to explain Figure 2 and Figure 3 by incorporating the references to D/P/N/E inside Figure 2 and 3 explicitly. For example, “we chose DP as the unified instruction format” would seem to make more sense if Figure 2 had “Definition” and “Positive Examples” highlighted accordingly. As of now, I am struggling to see how the specifically defined D/P/N/E fields flow into exemplars that are fed into GPT-3.5 which are then fed into training or testing. It also seems difficult to understand the paper without reading the Ni-v2 work first.
    * In the Ni-v2 work, the explanation seems to be a field of the positive (P) and negative (N) examples, but the current framing puts Examples (E) in the same logical level as D/P/N (e.g., “siblings” rather than “children”). This is potentially confusing, because P and N each have their own E.
    * The unified approach as well as the heuristic could also be explained with a few examples. The examples in Figure 2, Figure 3, and appendix don’t seem to clearly state their relationship with the D/P/N/E framework.
    * What is the behavior (e.g., hallucinations) if more fields are used in the Unified Instruction Format than allowed by a source dataset (e.g., if some of the fields in D/P/N/E are missing)? Section 6 mentions intentionally going from complex formats to simpler ones, which seems related.
    * What makes training so different from testing, given the differences noticed in the results?
    * Table 6 shows results generalizing past Ni-v2. Can you add an additional dataset other than flan?
* Some of the hyperparameters aren’t motivated. What is the significance of the four unified instruction formats selected (e.g., Table 2)? One of the observations is that DPN performed the best, but do we have any explanation for why this may be the case?
    * Can you provide examples of samples from Table 2 (with analysis) so the reader gains intuition?
* Cost for Section 6 is not motivated. Can you quantify how many LLM queries are required to train a model (Section 6)?

Nits:
* Table 1 seems squished due to abbreviations, but it’s still not clear what the reader is supposed to learn from it. The “P3” with “Total Exp. Entry” entry is inconsistently italicized.
* Table 2 “Average” column should be marked as different from the other columns representing datasets.
* Figure 2 has “theinformal” as one word.
* Some typos: “hypothsis”, “noises”
* “Without loss of generality” shouldn’t be used unless the results really hold in all of generality.

---

> ### Author Response · Authors · 2023-10-15
> **Responses to Reviewer nGZc (Part I)**
>
> Thank you for the comprehensive feedback on our manuscript. Below, we have provided detailed responses to each of your remarks and inquiries.
>
>
>
> **Regarding Weakness 1**
>
> - **Structure**: We appreciate the suggestion to restructure our paper for enhanced readability. We will reorganize the structure in future revisions.
> - **Overlap**: Regarding Table 5, we believe that the overlapping data offers clarity and is pivotal for comparisons between the unified method and the denoised version. Its inclusion serves both utility and comprehension for our audience.
> - **Section 7 Coherency**: We admit that these aspects in section 7 lack strong inherent connections, and we have organized them as individual takeaways to gain a more comprehensive understanding of the overall approach.
>
>
>
> **Regarding Weakness 2**
>
> - **Training Details**: We didn't do pre-training for the models, instead we used the trained models from huggingface, and we fine-tune the model with the transfered data in the training time and testing time setting. Our difference with Tk-instruct is that we fine-tune on our unified format. More training details about the models are below:
>
>   >  max_source_length = 1024; max_target_length = 128; max_num_instances_per_task = 100; max_num_instances_per_eval_task = 100; batch size = 16; learning_rate = 1e-05; num_train_epochs = 2; lr_scheduler_type = linear; warmup_steps = 1000
>
> - **Manual Labelled Data**: We have added the manually labelled seed data to the appendices.
>
> - **Heuristic:** Following is the definition of our heuristic method. When information from the original format aligns with a field in our proposed format, we populate that field accordingly. If not, the field remains empty. Here is an example: A task from Flan reading "You need to complete..." would be transformed into Ni-v2's DP format as:
>
>   ```
>   Task description: You need to complete...
>
>   Positive Example: None.
>   ```
>
>   Our Unified Instruction Tuning method will be more convinient when the format is complicated.
>
>
>
> **Regarding Weakness 3**
>
> - **Specific Results**: We have added the specific results to the abstract and conclusion.
> - **Readability**: We have (i) convert the original Table 4 and Table 6 into line plots and bar plots; (ii) mark the 'Avergae' column with different color to enhance readability. We will continue to improve this.
>
>
>
> **Regarding Weakness 4**
>
>
>
> - **About the Abbreviation**: Regarding the 'E' abbreviation, it represents 'Explanations' and not 'Examples'. As highlighted in Section 4.2 under the 'Datasets' paragraph, we clarified the DPNE terminology in relation to Ni-v2. Throughout the paper, the abbreviations D, P, N, and E remain consistent with this.
>
> - **About the Examples:** Figure 2 specifically illustrates the 'D' format transformation, which pertains solely to task description alterations. The transformations for PNE can be referenced in the examples provided in our appendices. We have also added more examples of the approach in the appendices.
>
> - **About More Fields**: In our experiments, `text-davinci-003` demonstrated an ability to manage additional fields. We explored adding fields such as 'Chain of Thought' and 'Constraints', and observed promising outputs. However, preliminary results indicated that these additions weren't significantly impactful. Our ongoing work aims to harness this data more effectively, particularly for elements like 'Chain of Thought'. Additionally, to address potential inaccuracies in outputs from `text-davinci-003`, we've integrated a denoising technique.
>
> - **About Setting Difference**: With respect to test settings, Ni-v2 was selected as the primary training dataset, while DiversePrompt, Flan, CrossFit, and PromptSource served as testing datasets. For training scenarios, Ni-v2, along with other datasets, were utilized for training and Ni-v2 was retained for testing.
>
> - **About Additional Dataset in Table 6**: We've previously established the efficacy of UIT across various datasets with Ni-v2 as the reference. The objective in transitioning Ni-v2 to Flan is to validate that UIT remains effective, even when shifting target formats. Furthermore, to our understanding, the formats of CrossFit and p3 are closely aligned with Flan.

---

> ### Author Response · Authors · 2023-10-15
> **Responses to Reviewer nGZc (Part II)**
>
> **Regarding Weakness 5**
>
> - **About why we choose these four formats**:  We selected four combinations of instruction formats based on human experience and previous research, such as Ni-v2 and Flan. From our hypothesis, we understood that 'D' is fundamental and necessary, while 'P,' 'N,' and 'E' would likely decrease in importance. Therefore, we opted for the following four combinations: 'DP,' 'DPN,' 'DPE,' and 'DPNE.' Our objective is not to identify the most ideal format for all scenarios; instead, we aim to demonstrate that transitioning to a specific format can enhance our performance.
> - **About why DPN performs the best**: Our hypothesis is that the negative examples will directly give the bad cases that is helpful for model improvement, and 'explanation' part is not so informative comparing with 'negative' in our cases.
> - **About Examples from Table 2:** We have added some examples of DPNE in the appendices.
>
>
>
> **Regarding Weakness 6**
>
> - **About query amount and cost**: We utilized approximately 3,000 GPT-3.5 queries to gather the training data for the model. This incurred a cost of roughly 72 dollars at a rate of 0.024 dollar per query, considering 900 prompt tokens and 300 response tokens.
>
>
>
> **Regarding Nits**
>
> Thank you for the nits found. We have fixed them up in the revised version of our paper.
>
>
>
> **We greatly appreciate your feedback and have made revisions to the article based on your suggestions.**

---

### Comment · Action_Editors · 2023-12-20
**Revise the format.**

Dear authors,

The section 'Appendices' is usually put after 'References'.  You can refer the author guidance (https://www.jmlr.org/tmlr/author-guide.html) to revise the format.

Thanks.

---

> ### Author Response · Authors · 2023-12-20
>
> Thank you so much for your feedback!
>
> We've followed the instructions and adjusted the order of the 'Appendices' and 'References' sections in our manuscript.
>
> Best regards.

---

### Decision · Action_Editor_VpVy · 2023-11-20

**Recommendation:** Accept as is

**Comment:**

This paper models and discusses format consistency for instruction tuning, which is important to LLM-based applications. The focused problem is interesting and comprehensive quantitative results verify the effectiveness of the method. Though reviewers raise some concerns (e.g., the method's motivation) , most of them have been resolved in n the response phase. All the reviewers agree that the proposed method fit the TMLR acceptance criteria and post positive comments.

Overall, I recommend accepting this paper.

**Audience:**

Yes. Some individuals in the TMLR community will be interested in this paper.

**Claims And Evidence:**

Yes. The authors conduct extensive experiments to support the claims.